# EFFICIENT RECURRENT NEURAL NETWORKS USING STRUCTURED MATRICES IN FPGAS

**Zhe Li[1], Shuo Wang[2], Caiwen Ding[1], Qinru Qiu[1], Yanzhi Wang[1] & Yun Liang[2]**
[1]Department of Electrical Engineering and Computer Science, Syracuse University, USA
[2]Center for Energy-efficient Computing and Applications (CECA), Peking University, China
[1]{zli89, cading, qiqiu, ywang393}@syr.edu
[2]{shvowang, ericlyun}@pku.edu.cn

## ABSTRACT

Recurrent Neural Networks (RNNs) are becoming increasingly important for time series-related applications which require efficient and real-time implementations. The recent pruning based work *ESE* (Han et al., 2017) suffers from degradation of performance/energy efficiency due to the irregular network structure after pruning. We propose block-circulant matrices for weight matrix representation in RNNs, thereby achieving simultaneous model compression and acceleration. We aim to implement RNNs in FPGA with highest performance and energy efficiency, with certain accuracy requirement (negligible accuracy degradation). Experimental results on actual FPGA deployments shows that the proposed framework achieves a maximum energy efficiency improvement of $35.7\times$ compared with ESE.

## 1 INTRODUCTION

Hardware implementations and model compression of Recurrent Neural Networks (RNNs) exhibit unique challenges. RNNs are very sensitive to accumulation of imprecisions, due to both model compression and bit quantization. This is because RNNs are equivalent to an infinite-depth neural network in which imprecision accumulation is more significant compared with finite-depth neural networks (Han et al., 2017; Liao et al., 2017; Han et al., 2015a). As a representative work on implementing LSTMs on FPGAs, the ESE (Han et al., 2017) implements sparse LSTM model obtained by the parameter pruning method (Han et al., 2015a;b). The ESE achieves higher energy efficiency than GPU, but its performance is lower and it cannot operate in real time. This is due to (i) the limited compression ratio for LSTMs ($4\text{-}6\times$ when indices are accounted for), (ii) the irregular network structure after pruning, and (iii) the inefficient implementation of activations and indices.

We propose to apply block-circulant matrix, a structured matrix, to RNNs, which significantly reduces computational and storage complexity and becomes amenable to hardware. It can also overcome the irregularity issue in RNN hardware implementation of the prior weight pruning methods.

## 2 BLOCK-CIRCULANT MATRIX REPRESENTATION

The primary idea of block-circulant matrix-based RNN is to represent the original arbitrary-size weight matrix $\mathbf{W} \in \mathbb{R}^{m \times n}$ with an array of equal-size square sub-matrices (i.e., *blocks*), where each sub-matrix is a *circulant* matrix. Assume there are $p \times q$ blocks after partitioning the matrix $\mathbf{W}$, where $p = \frac{m}{k}$ and $q = \frac{n}{k}$. Here $k$ is the *block size*. Then $\mathbf{W} = [\mathbf{W}_{ij}]$, $i \in \{1 \ldots p\}$, $j \in \{1 \ldots q\}$.

Each circulant matrix $\mathbf{W}_{ij}$ can be defined by a vector $\mathbf{w}_{ij}$. More specifically, $\mathbf{w}_{ij}$ is the first row vector of $\mathbf{W}_{ij}$; the second row vector of $\mathbf{W}_{ij}$ is a circulation of the first row vector, and so on. Figure 1 provides an example of circulant matrix $\mathbf{W}_{ij}$. The storage complexity of a block-circulant weight matrix is significantly reduced since we only need to store one vector $\mathbf{w}_{ij}$ for each circulant matrix $\mathbf{W}_{ij}$. As a result, we have the ability to store all the weights matrices in block RAM (BRAM), thereby significantly improving the FPGA performance. Additionally, the input feature $\mathbf{x}$, bias $\mathbf{b}$ can also be stored in BRAM due to a small quantity of corresponding parameters.

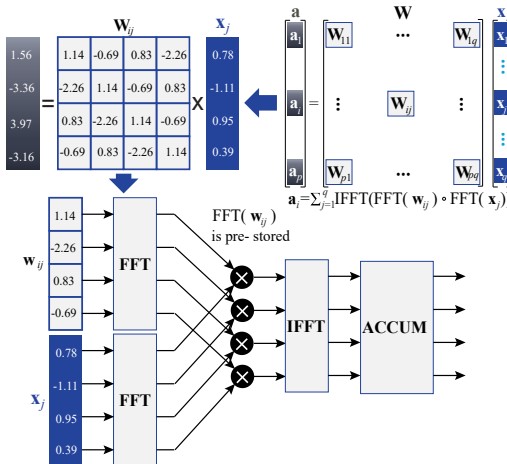

Figure 1: An illustration of FFT-based calculation in block-circulant matrix multiplication.

Since a weight matrix $\mathbf{W}$ is now partitioned into $p \times q$ blocks, correspondingly, the input $\mathbf{x}$ is also partitioned as $\mathbf{x} = [\mathbf{x}_1^T, \mathbf{x}_2^T, \ldots, \mathbf{x}_q^T]^T$, $\mathbf{x}_j \in \mathbb{R}^k$. Then, the **forward propagation** process in the inference phase is given by (with bias and activation function omitted):

$$\mathbf{a} = \mathbf{W}\mathbf{x} = \begin{bmatrix} \sum_{j=1}^q \mathbf{W}_{1j}\mathbf{x}_j \\ \sum_{j=1}^q \mathbf{W}_{2j}\mathbf{x}_j \\ \cdots \\ \sum_{j=1}^q \mathbf{W}_{pj}\mathbf{x}_j \end{bmatrix} = \begin{bmatrix} \mathbf{a}_1 \\ \mathbf{a}_2 \\ \cdots \\ \mathbf{a}_p \end{bmatrix}, \tag{1}$$

where $\mathbf{a}_i \in \mathbb{R}^k$ is a column vector. We can see the calculation of $\mathbf{W}\mathbf{x}$ is reduced to the calculation of $\mathbf{W}_{ij}\mathbf{x}_j$'s. Then according to (Pan, 2012; Bini et al., 1996), the calculation of $\mathbf{W}_{ij}\mathbf{x}_j$ can be performed as

$$\mathbf{W}_{ij}\mathbf{x}_j = \mathbf{IFFT}\big(\mathbf{FFT}(\mathbf{w}_{ij}) \odot \mathbf{FFT}(\mathbf{x}_j)\big), \tag{2}$$

where $\odot$ denotes element-wise multiplications, and **FFT** and **IFFT** denote Fast Fourier Transform (FFT) and inverse FFT, respectively. The computational complexity of $\mathbf{W}\mathbf{x}$ is reduced from $O(n^2)$ by direct matrix-vector multiplication to $O(pqk \log k)$ by the "FFT→element-wise multiplication→IFFT" procedure in Equation (2), which is equivalent to $O(n \log n)$ for small $p$, $q$ values. As a result, the simultaneous acceleration and model compression compared with the original RNN can be achieved for the inference process.

The **backward propagation** process in the training phase can also be implemented using block-circulant matrices, which is similar to the procedure in (Ding et al., 2017). It is important to understand that during training, the block-circulant matrix-based approach directly trains weight matrices in the block-circulant format by training only one vector for each block (i.e., circulant matrix). In other words, **there is no need for the re-training process**. This is distinctive compared with other prior works (Han et al., 2015a;b) which increase the training complexity due to the additional pruning and re-training processes.

## 3 LSTM MODEL EXPLORATION AND HARDWARE RESULTS ON FPGA

We explored the LSTM model on TIMIT dataset (Garofolo et al., 1993) for different configurations in Table 1. For an LSTM cell, $1024-1024$ means that the network has two layers of LSTM cells with 1024 hidden neurons. The block sizes are listed in the same format as layer sizes correspondingly. "$-$" means no circulant matrix applied on the network, serving as the **the baseline model**. We also applied "peephole" and the "projection" layer of 512 to the LSTM models (Sak et al., 2014). The models has the exact same network architecture as ESE in (Han et al., 2017). The performance is evaluated by *phone error rate* (PER) and degradation compared to the corresponding baseline model.

From the Table 1, we can observe that the block-circulant matrix-based framework results in very small accuracy degradation compared with the baseline model. More specifically, when the block

Table 1: Accuracy Comparison among LSTM based RNNs on TIMIT

| ID | Layer Size | Block Size | Phone Error Rate (PER) % | PER degradation (%) |
|----|-----------|-----------|--------------------------|---------------------|
| 1 | $1024 - 1024$ | – | 20.01 | – |
| 2 | $1024 - 1024$ | $4 - 4$ | 20.01 | 0.00 |
| 3 | $1024 - 1024$ | $4 - 8$ | 20.05 | 0.04 |
| 4 | $1024 - 1024$ | $8 - 4$ | 20.10 | 0.09 |
| 5 | $1024 - 1024$ | $8 - 8$ | 20.14 | 0.13 |
| 6 | $1024 - 1024$ | $8 - 16$ | 20.22 | 0.21 |
| 7 | $1024 - 1024$ | $16 - 8$ | 20.29 | 0.28 |
| 8 | $1024 - 1024$ | $16 - 16$ | 20.32 | 0.31 |

Table 2: Detailed comparisons for different LSTM designs on FPGAs (ours and ESE).

| | ESE (Han et al., 2017) | Block-Circulant RNN FFT8 (Block size: 8) | | Block-Circulant RNN FFT16 (Block size: 16) | |
|---|---|---|---|---|---|
| Matrix Size (#Params of top layer) | 0.73M | 0.41M | | 0.20M | |
| Model Compression Ratio | $4.5 : 1$[a] | $7.9 : 1$ | | $15.9 : 1$ | |
| Platform | KU060 | KU060 | 7V3 | KU060 | 7V3 |
| PER Degradation | 0.30% | 0.14% | | 0.31% | |
| Latency ($\mu$s) | 57.0 | 13.7 | 12.9 | 7.4 | 8.3 |
| Frames per Second (FPS) | 17,544[b] | 231,514 | 240,389 | 429,327 | 382,510 |
| Power (W) | 41 | - | 24 | - | 25 |
| Energy Efficiency (FPS/W) | 428 | - | 10,016[c] | - | 15,300[c] |

[a] This estimation considers both weights and indices (there is at least one index per weight after compression in ESE). However, this is a pessimistic estimation for ESE because indices can use fewer bits for representation than weights.
[b] We use ESE's theoretical computation time to calculate its FPS, the real computation time is larger than the theoretical one which leads to a smaller FPS.
[c] The resource of the FPGA chip Virtex-7 of ADM-7V3 platform is 30% higher than the FPGA XCKU060 of KU060 platform. To make a fair comparison, we use the total resource of KU060 as the resource consumption bound for AMD-7v3 platform.

size is 4 (4 × parameter reduction), there is no accuracy degradation compared with the corresponding baseline. When the block size is 8 (8 × parameter reduction), the accuracy degradation is negligible. When the block size is 16, the accuracy degradation is still only around 0.3%. We can conclude that the block-circulant matrix-based framework outperforms ESE in terms of model compression. This is because ESE achieves 8.9× parameter reduction (This parameter reduction does not account for the indices needed for indexing, which are at least one for each parameter in the network structure after pruning) with 0.3% accuracy degradation. In our experiment No.6 and No.7, we have more parameters reduced and better accuracies than ESE.

We observe in the hardware experimental results in Table 2 that the Frame Per Second(FPS) and energy efficiency gains are even more significant compared with ESE. As the regularity in our proposed framework architecture results in a high degree of parallelism. We use two FPGA platforms for evaluating the proposed Block-Circulant LSTM: Alpha Data's ADM-PCIE-7V3 and Xilinx KU060. The proposed model's FPGA implementation is operating at 200MHz on both platforms, which is configured to be the same as the prior work ESE Han et al. (2017) for fair comparisons. We both used 12bit fixed-point number to represent the values. We present a direct comparison between our block-circulant RNN and ESE with the same baseline LSTM model with layer size 1,024.

In the first case, the block size is 8 and the compression ratio is 7.9×. The comparison results, as shown in the first two columns of Table 2, are both on the KU060 FPGA platform. We could observe that the our model achieves lower accuracy degradation compared with ESE (0.14% vs. 0.30%), demonstrating the effectiveness of the block-circulant framework in terms of accuracy. Meanwhile, we can observe that our model achieves 13.2× FPS improvement with a block size of 8, with an energy efficiency improvement of 23.4× using actual measurement results on the ADM-PCIE-7V3 board. It is necessary to note that, the manufacture process of XCKU060 FPGA is 20nm while the process of Virtex-7 is 28nm, which means the energy efficiency gain reported here is even conservative. In the second case, the block size is 16 and the compression ratio is 15.9×. Our model achieves similar accuracy but 24.5× FPS improvement and 35.7× energy efficiency improvement.

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
