# OpenReview forum: "Efficient Recurrent Neural Networks using Structured Matrices in FPGAs"
_ICLR.cc/2018/Workshop — Accept_

### Official Review · AnonReviewer1 · 2018-03-09
**More details are needed**

**Rating:** 7
**Confidence:** 4

**Review:**

This paper targets at compressing RNN models to reduce model size, computation cost and enhance model efficiency, without  too much performance drop. This paper proposes to utilize block circulant matrix to speed up the computation as BC matrix multiplication can be computed fast in FFT domain.

The idea is straightforward. It seems similar idea has been explored for compressing CNN models  (Ding et al., 2017). But this should be the first work applying BC matrix to compress RNN models. The performance is much better than ESE, in terms of both model computation efficiency and performance in speech applications.

The paper is written well. The only issue is it is not clear how BC matrix can be used to compress LSTM, considering LSTM has several different gates with complex interaction. I understand the space is limited but would appreciate if the authors would provide more implementation details for LSTM.

---

### Official Review · AnonReviewer3 · 2018-03-10
**interesting work, though results would be strengthened if reported on a larger task (e.g., Switchboard)**

**Rating:** 6
**Confidence:** 3

**Review:**

Overall, this is an interesting paper which discusses techniques to build compact and efficient RNN models.
The work presented is similar to work presented previously, and I think the authors should list the following paper and discuss similarity/differences with this work: (Sindhwani et al., 15)
V. Sindhwani, T. Sainath, S. Kumar, “Structured Transforms for Small-footprint Deep Learning,” Neural Information Processing Systems (NIPS), 2015.
Ideally, it would be interesting to compare results with the proposed techniques on a larger ASR task with more data than TIMIT, e.g., Switchboard. While the TIMIT results are certainly compelling, it would be more interesting to consider performance on a larger word recognition task. Another related question: What was the variance in performance across runs?
Minor comment: Section 2: “The computational complexity is reduced from O(n^2)”. Should this be O(mn)?
Minor comment: ESE - Please explain what this acronym is in the text.

---

### Official Review · AnonReviewer2 · 2018-03-11
**This paper is about exploiting block-structured matrices for efficient implementation of RNN in FPGA. Good paper. The source code should be shared.**

**Rating:** 7
**Confidence:** 3

**Review:**

This paper is about exploiting block-structured matrices for efficient implementation of RNN in FPGA. Previous work have showed  that implementing  LSTM in FPGA  achieves higher energy efficiency than GPU, but performs poor in accuracy. This work proposes to overcome this by relying on block-structured matrices.

The paper is well written. Of course, several parts need to be detailed, but as a three pages workshop paper the essence is there. However I highly recommend to the authors to share their code so as the community could regenerate the empirical results. This is mandatory in my opinion for the paper to be accepted.

---

### Decision · Program_Chairs · 2018-03-20
**ICLR 2018 Workshop Acceptance Decision**

**Decision:**

Accept

**Comment:**

Congratulations, your paper was accepted to the ICLR workshop.